# The experiences of trained breastfeeding support providers that influence how breastfeeding support is practised: A protocol for a qualitative evidence synthesis

**Mary Jo Chesnel** *, **Maria Healy**, **Jenny McNeill**

School of Nursing and Midwifery, Medical Biology Centre, Queen's University Belfast, Belfast, Northern Ireland

* mchesnel01@qub.ac.uk

## Abstract

### Background

Many women stop breastfeeding before they intend to as they cannot overcome breastfeeding difficulties. Breastfeeding support, as an evidence-based intervention by trained lay or professional breastfeeding support providers, can prevent early unintended cessation. Yet some women report dissatisfaction with support and reluctantly stop breastfeeding despite receiving this intervention. Understanding the experiences which shape how support is provided can inform effective implementation of breastfeeding support interventions. This review aims to synthesise experiences of trained breastfeeding support providers in high income settings and how these may influence their breastfeeding support practices.

### Methods

A qualitative systematic review of trained breastfeeding supporters' experiences of supporting women to breastfeed, as part of a generic healthcare role or focused breastfeeding support role, will be conducted. A systematic search will be performed of the databases: Cumulative Index to Nursing and Allied Health Literature (CINAHL +), MEDLINE ALL, Maternity and Infant Care, EMBASE, APA PsycINFO, Web of Science and Scopus. Title and abstract screening using eligibility criteria will be conducted using *Covidence* software. Eligible papers will be agreed by the review team following full text screening and reported using PRISMA guidelines. CASP and COREQ tools will assess study methodological quality and quality of reporting. Data will be extracted using a bespoke form and coded, using Excel software for data management. Analysis will involve the three stages of thematic synthesis: initial free coding, development of descriptive and subsequent analytical themes. Confidence in findings will be assessed using the CERQual framework.

### Discussion

This review is the first to date to synthesise qualitative evidence on experiences which influence how trained lay and professional providers support women with breastfeeding.

**Funding:** This review is funded as part of the Department of the Economy (Northern Ireland) funding of Doctoral training for the first author MJ Chesnel. The funder had no role in the design of this study protocol, decision to publish or production of the protocol manuscript. They will have no role in data collection, analysis, decision to

publish or preparation of the subsequent review manuscript.

**Competing interests:** No authors have competing interests.

**Abbreviations:** APA PsycINFO, American Psychological Association PsycINFO; CASP, Critical Appraisal Skills Programme; CERQual, Confidence in the Evidence from Reviews of Qualitative Research; CINAHL, The Cumulative Index to Nursing and Allied Health Literature; COREQ, Consolidated Criteria for Reporting of Qualitative Research; EMBASE, Excerpta Medica Database; MeSH, Medical Subject Headings; PEOT, Population, Exposure, Outcome, Type of study; PRISMA, Preferred Reporting Items for Systematic Reviews and Meta-analyses; PROSEPERO, The International Prospective Register of Systematic Reviews.

Findings will enable deeper understanding of the underpinning mechanisms of breastfeeding support provision and inform the development of tailored interventions to improve breastfeeding rates.

## Systematic review registration

PROSPERO registration number: CRD42020207380

## Introduction

The benefits of breastfeeding and associated risks of not breastfeeding have been widely reported in the literature [1–3]. Breastfeeding has been identified as crucial in meeting the United Nations (UN) Sustainable Development Goals for 2030 [4] with the World Health Organisation aiming for global rates of 50% exclusive breastfeeding until 6 months of age by 2025 [5]. Despite the scientific evidence of benefit, and global directives, breastfeeding rates remain low in comparison to these recommendations, especially in high income countries [2, 3, 6]. Of women who start to breastfeed, many cease before they intended to do so citing challenges such as physical pain [7], perceived insufficient milk supply [8] (8) and breastfeeding not fitting in with family and/ or work life [9]. Such challenges are often surmountable with effective breastfeeding support [10–12].

Trained breastfeeding support providers, whether lay or professional, can prevent early unintended breastfeeding cessation [3, 13, 14]. Breastfeeding support is a complex intervention including sharing of advice and information, providing skilled help, reassurance and building the mother's confidence [15]. A wide variety of breastfeeding support-roles exist which require the support-provider to undergo breastfeeding training. A range of healthcare staff provide support as part of their role in the postnatal period, in maternity units and primary care settings. Trained volunteers support women to breastfeed in hospital and community settings. International board-certified lactation consultants are breastfeeding supporters employed in acute settings or private practice. Women seek organised or professional support when experiencing breastfeeding challenges [16, 17]. However, women also report different perceptions of support received including varying levels of satisfaction, and subsequent motivation and confidence to breastfeed [17–19]. There is known variance in the success of support interventions in terms of breastfeeding outcomes, for example, rates of duration and exclusivity [12, 20, 21]. Ultimately, breastfeeding support which is not effective or indeed, absent, can lead to women feeling as if they have failed at what is perceived to be a natural skill [22] and their breastfeeding experience sabotaged [23].

Extensive research into breastfeeding support has resulted in several systematic reviews focusing on breastfeeding support interventions. These reviews examine effectiveness in terms of breastfeeding rates [3, 6, 12, 20, 24–32], explore perceptions of breastfeeding support [33, 34] and investigate the use of theory in intervention design [35]. Most research focuses on the specifics of breastfeeding support intervention content, structure and settings, with the majority of studies offering a quantitative assessment of the effectiveness of interventions in terms of breastfeeding initiation, duration and exclusivity. Much less is known about the subjective and abstract factors that influence breastfeeding support provision. Little is known about the influencing factors subsequent to breastfeeding training on support provision, which impact on women's experience of receiving breastfeeding support. Two qualitative reviews exist, one exploring women's experiences of receiving breastfeeding support and one exploring midwives'

perceptions of their role in support provision [33, 34]. This aim of this review will differ in population and outcome from the review which explores midwives' perceptions of their role [33] as the population in this study is a range of trained breastfeeding support providers and the outcomes are the experiences which impact provision of that support. Undertaking a review which systematically explores the experiences that influence how breastfeeding support is provided, across a range of trained breastfeeding support provider roles, will result in synthesised qualitative evidence on breastfeeding support provision. The resultant body of evidence has the potential to inform the effective design and implementation of breastfeeding support interventions, drive future strategic policy, and improve care for breastfeeding women.

The search strategy is informed by the PEOT (Population, Exposure, Outcomes, Type) question format for qualitative research [36] with the following elements: P: Trained breastfeeding support providers, E: Breastfeeding support provision, O: Experiences that influence breastfeeding support practices, T: Studies with qualitative findings. The research questions are:

1. What is known about trained breastfeeding support providers' experiences that influence their provision of support?

2. How do trained breastfeeding support providers' personal and vicarious experiences of both breastfeeding and breastfeeding support provision influence their support practices?

3. Which support providers' experiences facilitate or impede their provision of breastfeeding support to women?

## Methods

This systematic review is registered with the International Prospective Register of Systematic Reviews (PROSPERO): Registration number CRD42020207380.

### Aim

This synthesis of qualitative research aims to identify, describe, and interpret qualitative research findings relating to the experiences of trained breastfeeding support providers in high income settings and how these may positively or negatively influence their breastfeeding support practices.

**Definition.** The term "breastfeeding support" in this review refers to proactive or reactive interactions between women, infants and trained breastfeeding support providers offering reassurance, praise, skilled help, problem solving, information and social support in face-to-face, group or digital settings such as social media groups, telephone calls or text messages. Support may be provided in acute hospital, maternity units, primary care, voluntary and community settings and women's own homes. This definition is adapted from McFadden *et al.* [3].

### Search strategy

Seven databases will be searched for eligible studies: CINAHL +, MEDLINE ALL, Maternity and Infant Care, EMBASE, APA PsycINFO, Web of Science and Scopus. A search strategy using MeSH headings, related keywords and truncations will be developed (see Table 1). The Boolean terms OR and AND will be used. Reference lists of retrieved eligible studies and the reference lists of unpublished literature sourced via Open Grey and British Library Ethos will be hand searched for relevant published studies. The search period will be from year 2003 – current. The start year of 2003 is chosen in order to identify research undertaken following publication of the World Health Organisation's Global Strategy for Infant and Young Child Feeding [37] which advised that women exclusively breastfeed for 6 months and continue

**Table 1. Search strategy.**

| Population: | Exposure: | Outcome: | Type of study: |
|---|---|---|---|
| Trained breastfeeding support providers | Breastfeeding support provision | Experiences that influence breastfeeding support practices | Studies with qualitative methods and findings |
| Limited to English language and year of publication 2003 –current | | | |
| Midwifery/OR Nurse midwives/OR Midwi\*.mp OR Health Personnel/ OR Physicians/ OR Doctors.mp OR Health visitors or nurses, community health/ OR Peer counselling.mp OR Volunteers/ OR Lactation Consultants.mp OR Breastfeeding counsellors. mp OR Breastfeeding supporters.mp OR Doulas.mp or Doulas/ | Breastfeeding promotion.mp OR Breastfeeding support.mp OR Breastfeeding.mp or Breastfeeding/ OR Lactation management.mp OR Infant Feeding.mp OR Breastfeeding Counseling | Experiences.mp perception\*.mp OR views.mp OR Feelings.mp or Emotions/ | Qualitative Research/or qualitative.mp OR Interviews.mp OR Focus groups.mp OR "mixed method\*" OR Ethnography OR Participant observation |
| Four PEOT search terms will be combined with **AND** | | | |

breastfeeding for two years and beyond for optimal health benefits to mother and infant. This target involves the use of trained breastfeeding support providers. An English language restriction and a methodological filter for type of study will be included.

## Study identification and selection

**Inclusion criteria.** Qualitative studies will be included which focus on the experiences of trained breastfeeding support providers, and how these experiences influence breastfeeding support provision. Published qualitative studies and mixed methods studies with qualitative findings will be included. Studies are required to report ethical approval and demonstrate evidence of data to support findings. The population will comprise trained breastfeeding support providers who support breastfeeding women with healthy infants in high income countries. For the purpose of this review a trained breastfeeding support provider is defined as any trained healthcare staff working with breastfeeding women and healthy infants/children as part of their role, and any breastfeeding support providers such as volunteer breastfeeding supporters and lactation consultants, who have undertaken formal accredited training, in high income countries (as defined by the World Bank). Studies of the experiences and perceptions of provision of breastfeeding support interventions in acute hospital, maternity units, primary care, voluntary and community settings and women's own homes will be included.

**Exclusion criteria.** Mixed-methods studies will be excluded if the qualitative findings are not reported. Students, untrained volunteers and healthcare staff working with sick infants/children are not included in the review as the focus is on routine breastfeeding support for healthy mothers with healthy babies. Breastfeeding support is not considered routine when delivered to women with additional care needs [38] or delivered in a neonatal or paediatric setting. Studies with heterogeneous samples including, for example, neonatal nurses or paediatricians will be excluded if data pertaining to the experiences of trained breastfeeding support providers working with women and healthy infants/children cannot be isolated from data from those working with sick children. Studies from low-income countries will be excluded.

**Data management.** Studies will be selected for inclusion following a two-stage process using *Covidence*, an online software programme designed to streamline and manage the systematic review process. Findings from the searches will be exported via EndNote X9 reference management software to *Covidence*. This will enable de-duplication of records and collaboration within the review team. The two-stage selection process will firstly involve screening of the title and abstracts by the first author MJC with verification by another independent reviewer (JM or MH). Disagreements will be resolved by discussion with a third reviewer (JM or MH). In the second stage of the selection process, all three authors (MJC, JM, MH) will independently

review each full text manuscript in detail. Manuscripts of all citations in reference lists of selected papers that are likely to meet the selection criteria will be retrieved by hand searching and assessed against the eligibility criteria. A flow diagram adapted from the Preferred Reporting Items for Systematic Reviews and Meta-Analysis (PRISMA) [39] guidance will be used to report the study selection process. The rationale for papers excluded will be reported.

A data extraction form will be developed and agreed by the team to extract data from the studies including: author, title, country, year, research aim, methodology, method, participant demographics. Most importantly, qualitative findings (themes and sub-themes) identifying experiences (for example, emotions, past encounters, training, practice) that influence how breastfeeding support is practiced, from the perspective of the trained breastfeeding support provider will be extracted as the main outcomes. A summary of each study's overall findings will be included in the form to give context to the data extracted.

## Study appraisal

The quality of included studies, assessed using the Critical Appraisal Skills Programme (CASP) [40] and the reporting of each study, assessed using the Consolidated Criteria for Reporting Qualitative Research Tool (COREQ) [41] will be assessed by the first author MJC and reviewed by JM and MH, and will be agreed by consensus. Studies will not be excluded based on the CASP assessment or COREQ score, rather the CASP and COREQ assessments will provide information to assist in assessing the credibility of the findings of each study, and subsequently inform a later assessment of confidence in the review findings using the CERQual approach [42].

## Data analysis and synthesis

A systematic three step process of line-by-line coding, generation of descriptive themes and subsequent generation of analytical themes will be undertaken in the synthesis. Coding will be conducted primarily by MJC and reviewed by the other authors with agreement by consensus. The stages of the thematic synthesis method will be used following guidance by Thomas and Harden [43]. An inductive approach to coding will be used. Firstly, all text in the findings sections of the included papers will be coded line-by-line, including both data from the study and the author's interpretations. These codes will be transcribed into an Excel spreadsheet with the related data to enable searchability and a connection to the supportive quotes in the papers. Next, descriptive themes will be derived from the initial codes in an iterative process involving moving forward and back between the codes using principles of thematic analysis [44]. Lastly, interpretation of the descriptive themes will lead to development of analytical themes that answer the research questions of the review, and the formulation of the thematic synthesis. Findings will be characterised across all papers and if possible a sub-analysis by role-type will be conducted.

## Confidence in review findings

The level of confidence in the review findings will be reported using the Confidence in Evidence from Review of Qualitative Research (CERQual) [45–47] approach. A CERQual Evidence profile will demonstrate whether the review authors have a high, moderate, low or very low confidence that an individual review finding is a reasonable representation of an experience which influences how breastfeeding support is provided to women.

## Discussion

Evidence-based breastfeeding support is an intervention that prevents early breastfeeding cessation yet little is known about the experiences of trained breastfeeding support providers who

implement the intervention, and whether such experiences influence how women are supported to breastfeed. This review will systematically collate, analyse, and synthesise the available evidence relating to experiences that influence how breastfeeding support is provided by trained breastfeeding support providers. This will provide synthesised qualitative evidence for breastfeeding support to inform effective breastfeeding support practice, education, future research and policy. This review is important as despite the abundance of evidence in relation to breastfeeding interventions, the influencing experiences (personal and professional) of breastfeeding supporters on their practice is not well understood. Evidence generation in the form of a qualitative synthesis will contribute to understanding the design and implementation of effective interventions seeking to impact on breastfeeding rates. Providing effective breastfeeding support will increase breastfeeding duration with significant impact on health and wellbeing of women and their families, and enable more infants to be exclusively breastfed until they are 6 months of age in line with the United Nations 2030 Sustainable Development Goals [4, 5].

## Supporting information

**S1 Checklist. The PRISMA-P 2015 checklist.**
(DOCX)

## Author Contributions

**Conceptualization:** Mary Jo Chesnel.

**Supervision:** Maria Healy, Jenny McNeill.

**Writing – original draft:** Mary Jo Chesnel.

**Writing – review & editing:** Mary Jo Chesnel, Maria Healy, Jenny McNeill.

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
