## [Decision Letter · Decision Letter 0]

7 May 2021

PONE-D-21-07631

The experiences of trained breastfeeding support providers that influence how breastfeeding support is practised: a protocol for a qualitative evidence synthesis.

PLOS ONE

Dear Dr. Chesnel,

Thank you for submitting your manuscript to PLOS ONE. After careful consideration, we feel that it has merit but does not fully meet PLOS ONE’s publication criteria as it currently stands. Therefore, we invite you to submit a revised version of the manuscript that addresses the points raised during the review process.

We look forward to receiving your revised manuscript.

Kind regards,

Jennifer Yourkavitch

Academic Editor

PLOS ONE

Additional Editor Comments:

Please note that one of the reviewers is the Academic Editor for this submission.

Please respond to all reviewer comments.

Journal Requirements:

Reviewers' comments:

Reviewer's Responses to Questions

**Comments to the Author**

1. Does the manuscript provide a valid rationale for the proposed study, with clearly identified and justified research questions?

Reviewer #1: Partly

Reviewer #2: Yes

2. Is the protocol technically sound and planned in a manner that will lead to a meaningful outcome and allow testing the stated hypotheses?

Reviewer #1: Partly

Reviewer #2: Yes

3. Is the methodology feasible and described in sufficient detail to allow the work to be replicable?

Reviewer #1: Yes

Reviewer #2: Yes

4. Have the authors described where all data underlying the findings will be made available when the study is complete?

Reviewer #1: No

Reviewer #2: No

5. Is the manuscript presented in an intelligible fashion and written in standard English?

Reviewer #1: Yes

Reviewer #2: Yes

6. Review Comments to the Author

You may also provide optional suggestions and comments to authors that they might find helpful in planning their study.

Reviewer #1: This protocol outlines an interesting systematic review. The protocol has clearly been well-thought out. However, I have a few major comments, followed by some more minor considerations.

Main comments:

• Lines 110-113 – The PEOT format seems to be informing the search strategy, but I don’t see a strong connection between this format and the research questions. Perhaps just use PEOT to frame the search strategy.

• The research questions need to be clarified and specified. Lines 115 and 118 – Whose experiences are these questions referring to? Line 116 – What kind of experiences is this about? Their experiences while providing breastfeeding support or inclusive of other types of experiences as well? Line 116 – What kind of experiences is this about? Their experiences while providing breastfeeding support or something else?

• Section 213 – How are you developing the codebook (e.g., inductive, deductive, or what mix?)? Will all three reviewers be involved in coding? If yes, how will you ensure intercoder agreement? If using a purely inductive approach and more than one person is going to code, how will you come to a common codebook that is applied consistently across all texts?

• Lines 223-224 – Can you say a little more about what you will look for in the analysis? Will you seek to disaggregate the findings at all or are you just going to characterize findings across all the papers (e.g., look at findings by different regions, low or high income populations within the high income countries, experiences by type of provider)?

Other comments:

• Line 125 – Clarify that this is a synthesis of qualitative research. It currently reads like the synthesis itself will be qualitative.

• Lines 103-108 – I think it may be helpful to just incorporate this paragraph into the methods section so that you see all of the key definitions in one place. Also, need to align this definition with description provided in inclusion criteria (lines 166-169).

• Line 152 – Consider including breastfeeding counseling as a search term under E? Consider including ethnography and participant observation as search terms under T.

• Line 173 – How are you defining sick versus healthy mothers and infants? There may be health issues that infants or mothers have but breastfeeding support they receive is still routine. So what are the conditions/contexts in which breastfeeding support is not considered routine?

• Line 219 – This does not have bearing on review for this paper, but from an efficiency and quality control perceptive I’m wondering why you are using Excel. I would strongly suggest looking at qualitative management software for doing the line-by-line coding. I believe the major software (ATLAS.ti, NVivo, MAXQDA) have compatibility with EndNote to support coding for literature reviews. Dedoose is a low cost option but I don’t know that it has the reference software compatibility. ATLAS.ti Cloud is also low cost, but as a new product for ATLAS, I’m not sure they’ve built in the reference software yet.

Reviewer #2: This is an excellent protocol. You mentioned describing experiences in high-income countries. What happens if you find references for low-income countries? That was not listed as an exclusion criterion.

7. PLOS authors have the option to publish the peer review history of their article (what does this mean?). If published, this will include your full peer review and any attached files.

Reviewer #1: No

Reviewer #2: No

---

## [Author Response · Author response to Decision Letter 0]

11 Jun 2021

Level 7, Medical Biology Centre,

Queen’s University of Belfast,

97, Lisburn Road

Belfast BT97BL 

UK

Email: mchesnel01@qub.ac.uk

Associate Editor, PLOS One

Date of resubmission: 11/06/21

Dear Editor, 

Thank you for reviewing our submission titled “The experiences of trained breastfeeding support providers that influence how breastfeeding support is practised: a protocol for a qualitative evidence synthesis”. I am attaching a revised manuscript, with agreement from the co-authors, following changes suggested by the two reviewers. I would like to thank both reviewers for their helpful comments which we believe have strengthened the paper. 

The request for re-submission included specific comments from both reviewers alongside suggestions to improve the formatting of the document. Details in relation to our responses to the reviewers’ comments are provided here, and in table format in the uploaded document. Reviewers comments/ questions and our responses are listed below.

We hope this has answered all of the comments made and appreciate the review of our paper. If further clarity is required on any aspect we are happy to address and look forward to your response on our revised version. 

Yours sincerely, 

Mary Jo Chesnel 

COMMENT : Please ensure that your manuscript meets PLOS ONE's style requirements, including those for file naming. Please review your reference list to ensure that it is complete and correct. Firstly, style requirements have been reviewed and amendments made. The street name has been removed from the title page (line 11 revised manuscript), errors in referencing style have been rectified and file names have been submitted as per PLOS One style requirements. 

RESPONSE: During the revision process contributorship has been amended to reflect the nature of the authors’ contributions as suggested by the supervisory team of MH and JM. This amendment in line 17 of the revised manuscript now reads : The protocol was primarily conceived and prepared by MJC with review and editing contributions by MH and JM. These amendments have been discussed and approved by all co-authors. References to Dr. Jenny McNeill as JMcN have been amended to JM for consistency throughout the paper.

COMMENT: We note that you have stated that you will provide repository information for your data at acceptance. Should your manuscript be accepted for publication, we will hold it until you provide the relevant accession numbers or DOIs necessary to access your data. If you wish to make changes to your Data Availability statement, please describe these changes in your cover letter and we will update your Data Availability statement to reflect the information you provide. 

RESPONSE: In order to clarify the position on data access, you had kindly offered to rectify the Data Availability statement if changes were described in this letter. We agree that the original statement in response to the submission question on data availability stating ‘No data sets were generated or analysed during the current study protocol development. De-identified research findings will be publicly available when the study is completed and published’ be changed to read: No new data will be generated. Existing data referred to are in the public domain and will be referenced as this reflects more clearly the nature of the manuscript as a protocol for a synthesis of existing published evidence. The online submission portal for revisions has enabled the data availability field to be amended as such.

COMMENT: Does the manuscript provide a valid rationale for the proposed study, with clearly identified and justified research questions? Reviewer #1: Partly Reviewer #2: Yes 

RESPONSE: Reviewer 1 has indicated ‘partly’ and has requested clarification of the research questions in a later comment. The response to that request will be addressed later in this letter in order to keep the order of the comments as received. As there is no specific reference to address the ‘partly’ in relation to the rationale, and Reviewer 2 had answered ‘yes’, it was felt by all authors that no revision to rationale was required. 

COMMENT: Is the protocol technically sound and planned in a manner that will lead to a meaningful outcome and allow testing the stated hypotheses? Reviewer #1: Partly Reviewer #2: Yes

RESPONSE: Reviewer 1 has several suggestions for improvement of the protocol that are addressed further in this letter, please see below for detailed comment. 

COMMENT: Have the authors described where all data underlying the findings will be made available when the study is complete?

Reviewer #1: No Reviewer #2: No 

RESPONSE: Both reviewers strongly indicated that clarification of information about data access was required. As indicated in earlier in this letter, we agree and have revised the Data Availability statement to read No new data will be generated. Existing data referred to are in the public domain and will be referenced. Thank you for this suggestion.

COMMENT Reviewer 1: 

• Lines 110-113 – The PEOT format seems to be informing the search strategy, but I don’t see a strong connection between this format and the research questions. Perhaps just use PEOT to frame the search strategy. 

RESPONSE: Lines 104- 107 (revised manuscript) have been amended as advised and now read: The search strategy is informed by the PEOT (Population, Exposure, Outcomes, Type) question format for qualitative research (36) with the following elements: P: Trained breastfeeding support providers, E: Breastfeeding support provision, O: Experiences that influence breastfeeding support practices, T: Studies with qualitative findings

COMMENT Reviewer 1:

• The research questions need to be clarified and specified. Lines 115 and 118 – Whose experiences are these questions referring to? Line 116 – What kind of experiences is this about? Their experiences while providing breastfeeding support or inclusive of other types of experiences as well? Line 116 – What kind of experiences is this about? Their experiences while providing breastfeeding support or something else?

RESPONSE: The review questions have been revised as requested: Lines 109-114 (revised manuscript) now read: 

1. What is known about trained breastfeeding support providers’ experiences that influence their provision of support?

2. How do trained breastfeeding support providers’ personal and vicarious experiences of both breastfeeding and breastfeeding support provision influence their support practices? 

3. Which support providers’ experiences facilitate or impede their provision of breastfeeding support to women?

COMMENT Reviewer 1:

• Section 213 – How are you developing the codebook (e.g., inductive, deductive, or what mix?)? Will all three reviewers be involved in coding? If yes, how will you ensure intercoder agreement? If using a purely inductive approach and more than one person is going to code, how will you come to a common codebook that is applied consistently across all texts?

RESPONSE: This comment was helpful in improving our reporting of our method. Lines 191-193 in the revised manuscript lines have been amended to now read: In the second stage of the selection process all three authors (MJC, JM and MH) will independently review each full text manuscript in detail. 

Line 219-221 (revised manuscript) now reads: Coding will be conducted primarily by MJC and reviewed by the other authors with agreement by consensus. 

Line 222 (revised text) now reads: An inductive approach to coding will be used.

COMMENT Reviewer 1: 

• Lines 223-224 – Can you say a little more about what you will look for in the analysis? Will you seek to disaggregate the findings at all or are you just going to characterize findings across all the papers (e.g., look at findings by different regions, low or high income populations within the high income countries, experiences by type of provider)? 

RESPONSE: Thank you for your question, we are initially interested in the sample as a whole. Findings will be characterised across all papers so that experiences which influence the provision of support, in general, are identified. However if possible we will analyse by role type, in particular comparison of experiences which influence breastfeeding support provision between those working in healthcare roles and those in breastfeeding support-specific non-healthcare roles. The manuscript has been revised accordingly, line 230 now reads: Findings will be characterised across all papers and if possible a sub-analysis by role-type will be conducted. 

COMMENT Reviewer 1:

• Line 125 – Clarify that this is a synthesis of qualitative research. It currently reads like the synthesis itself will be qualitative

RESPONSE: Lines 121-124 (revised manuscript) has been amended to read: This synthesis of qualitative research aims to identify, describe, and interpret qualitative research findings relating to the experiences of trained breastfeeding support providers in high income settings and how these may positively or negatively influence their breastfeeding support practices.

COMMENT Reviewer 1:

• Lines 103-108 – I think it may be helpful to just incorporate this paragraph into the methods section so that you see all of the key definitions in one place. Also, need to align this definition with description provided in inclusion criteria (lines 166-169)

RESPONSE: Amended as advised, the definition of breastfeeding support has moved into the Methods section. It now aligns with the inclusion criteria, and lines 126 – 132 (revised manuscript) now read as follows:

Definition:

The term “breastfeeding support” in this review refers to proactive or reactive interactions between women, infants and trained breastfeeding support providers offering reassurance, praise, skilled help, problem solving, information and social support in face-to-face, group or digital settings such as social media groups, telephone calls or text messages. Support may be provided in acute hospital, maternity units, primary care, voluntary and community settings and women’s own homes. This definition is adapted from McFadden et al. (3). 

COMMENT Reviewer 1: 

• Line 152 – Consider including breastfeeding counseling as a search term under E? Consider including ethnography and participant observation as search terms under T.

RESPONSE: Line 153 (revised manuscript) Table 1 search terms amended as advised to include Breastfeeding counseling under E and Ethnography and Participant observation under T.

COMMENT Reviewer 1: 

• Line 173 – How are you defining sick versus healthy mothers and infants? There may be health issues that infants or mothers have but breastfeeding support they receive is still routine. So what are the conditions/contexts in which breastfeeding support is not considered routine?

RESPONSE: Thank you for highlighting this issue. In this review it is considered that although infants and mothers with health issues do receive breastfeeding support, supporting breastfeeding in contexts where the woman or infant requires additional care is no longer routine as the experience of support provision is altered in the face of additional concerns and care needs. This mirrors the 2017 Cochrane review which investigated “Support for healthy breastfeeding mothers with healthy term babies” (McFadden et al. 2017) establishing a baseline of evidence for breastfeeding support provision before any additional considerations. For the purpose of this paper, breastfeeding support is not considered routine when the mother has additional care needs or is being supported to feed her infant/child in a neonatal or paediatric setting. This aligns with the language of the UK Nursing and Midwifery Council midwifery proficiencies (2019) which refers to universal care for all women and additional care when required. Line 176 (revised manuscript) has been amended to read: Breastfeeding support is not considered routine when delivered to women with additional care needs (38) or delivered in a neonatal or paediatric setting. 

COMMENT Reviewer 1:

 • Line 219 – This does not have bearing on review for this paper, but from an efficiency and quality control perceptive I’m wondering why you are using Excel. I would strongly suggest looking at qualitative management software for doing the line-by-line coding. I believe the major software (ATLAS.ti, NVivo, MAXQDA) have compatibility with EndNote to support coding for literature reviews. Dedoose is a low cost option but I don’t know that it has the reference software compatibility. ATLAS.ti Cloud is also low cost, but as a new product for ATLAS, I’m not sure they’ve built in the reference software yet

RESPONSE: Thank you for your helpful advice and the use of management software will be considered.

COMMENT Reviewer 2:

• This is an excellent protocol. You mentioned describing experiences in high-income countries. What happens if you find references for low-income countries? That was not listed as an exclusion criterion.

RESPONSE: Thank you for your positive feedback. Studies from low-income countries will be excluded if found. The focus of this review is breastfeeding support in high-income countries as in most high-income countries the proportion of infants breastfeeding are lower than the global average (Victora et al. 2016). Within the exclusion criteria section of the revised manuscript, lines 181-182 have been amended to read: Studies from low-income countries will be excluded.

McFadden, A., Gavine, A., Renfrew, M. J., Wade, A., Buchanan, P., Taylor, J. L., Veitch, E., Rennie, A. M., Crowther, S. A., Neiman, S. & MacGillivray, S. 2017. Support for healthy breastfeeding mothers with healthy term babies. Cochrane Database of Systematic Reviews, 2, CD001141.

Nursing and Midwifery Council 2019. Standards of proficiency for midwives. London: Nursing & Midwifery Council.

Victora, C. G., Bahl, R., Barros, A. J. D., França, G. V. A., Horton, S., Krasevec, J., Murch, S., Sankar, M. J., Walker, N. & Rollins, N. C. 2016. Breastfeeding in the 21st century: epidemiology, mechanisms, and lifelong effect. The Lancet, 387, 475-490.

---

## [Editor Report · Decision Letter 1]

28 Jun 2021

The experiences of trained breastfeeding support providers that influence how breastfeeding support is practised: a protocol for a qualitative evidence synthesis.

PONE-D-21-07631R1

Dear Dr. Chesnel,

We’re pleased to inform you that your manuscript has been judged scientifically suitable for publication and will be formally accepted for publication once it meets all outstanding technical requirements.

Kind regards,

Jennifer Yourkavitch

Academic Editor

PLOS ONE
---

## [Editor Report · Acceptance letter]

14 Jul 2021

PONE-D-21-07631R1 

The experiences of trained breastfeeding support providers that influence how breastfeeding support is practised: a protocol for a qualitative evidence synthesis. 

Dear Dr. Chesnel:

I'm pleased to inform you that your manuscript has been deemed suitable for publication in PLOS ONE. Congratulations! Your manuscript is now with our production department. 

Kind regards, 

on behalf of

Dr. Jennifer Yourkavitch 

Academic Editor

PLOS ONE